# Mass Spectrometric-Based Proteomics for Biomarker Discovery in Osteosarcoma: Current Status and Future Direction

**DOI:** 10.3390/ijms23179741

**Published:** 2022-08-28

**Authors:** Nutnicha Sirikaew, Dumnoensun Pruksakorn, Parunya Chaiyawat, Somchai Chutipongtanate

**Affiliations:** 1Musculoskeletal Science and Translational Research (MSTR) Center, Department of Orthopedics, Faculty of Medicine, Chiang Mai University, Chiang Mai 50200, Thailand; 2Department of Biochemistry, Faculty of Medicine, Chiang Mai University, Chiang Mai 50200, Thailand; 3Center of Multidisciplinary Technology for Advanced Medicine (CMUTEAM), Faculty of Medicine, Chiang Mai University, Chiang Mai 50200, Thailand; 4Division of Epidemiology, Department of Environmental and Public Health Sciences, University of Cincinnati College of Medicine, Cincinnati, OH 45267, USA

**Keywords:** osteogenic sarcoma, proteome, mass-spectrometry, biomarker, therapeutic target

## Abstract

Due to a lack of novel therapies and biomarkers, the clinical outcomes of osteosarcoma patients have not significantly improved for decades. The advancement of mass spectrometry (MS), peptide quantification, and downstream pathway analysis enables the investigation of protein profiles across a wide range of input materials, from cell culture to long-term archived clinical specimens. This can provide insight into osteosarcoma biology and identify candidate biomarkers for diagnosis, prognosis, and stratification of chemotherapy response. In this review, we provide an overview of proteomics studies of osteosarcoma, indicate potential biomarkers that might be promising therapeutic targets, and discuss the challenges and opportunities of mass spectrometric-based proteomics in future osteosarcoma research.

## 1. Introduction

Osteosarcoma, also known as osteogenic sarcoma, is the most common type of primary malignant bone tumor [1,2]. Unlike other solid tumors, this cancer more frequently occurs in children and adolescents during their growth spurt, which is slightly higher in males than females [3]. The standard therapy comprises multi-agent neoadjuvant chemotherapy with doxorubicin, cisplatin, and often with high-dose methotrexate, followed by surgery and adjuvant chemotherapy. With this treatment, the five-year survival rate of osteosarcoma patients with localized disease is 65–70%. Unfortunately, the patient develops resistance to neoadjuvant chemotherapy, resulting in metastasis and a five-year survival rate of 15–30% [1,4]. Based on the clinical outcome of osteosarcoma patients without overt metastatic sites at diagnosis, 90% of patients developed lung metastasis 6–36 months later, but metastases can also develop in bone (8–10%) and rarely in lymph nodes. Most non-metastatic patients are suspected of having a micro-metastatic disease at the time of diagnosis [5,6]. Therefore, many attempts have been made to discover effective therapeutic targets, biomarkers for predicting chemoresistance, and monitoring indicators for early metastasis to improve the survival of osteosarcoma patients.

Owing to the completion of the human genome project, our knowledge of the genetic factors influencing cancer development has markedly improved. Proteomic studies are scarce relative to the abundance of genomic studies. Hence, the human cancer proteome remains underexplored [7,8]. Proteomics has provided complimentary and contrasting data to their genomic counterparts, leading to a comprehensive understanding of the molecular mechanisms underlining the pathology of diseases [9].

Mass spectrometric (MS)-based proteomics can be used in both discovery proteomics and targeted proteomics [10]. Discovery proteomics enables large-scale protein identification and the detection of protein dynamics in biological states and pathogenic conditions [11]. Targeted proteomics focuses on the precise detection and absolute quantitation of the selected target (i.e., peptides and their inferred proteins) in complex samples [12]. Current proteomics involves not only the study of protein abundance but also the analysis of protein regulation and activity and post-translational modifications (PTMs) of the proteins. Many enzymes and proteins involved in key signaling pathways are activated and deactivated through phosphorylation catalyzed by various kinases and phosphatases [13,14].

MS-based proteomics has emerged as the preferred approach for identifying and quantifying essential proteins in biological samples. This has considerably enhanced the unraveling of cellular signaling networks, the dynamics of protein–protein interactions, and a better understanding of molecular mechanisms under pathological conditions, which may eventually allow for personalized treatment [15,16]. This review summarizes the identification of biomarkers and novel therapeutic targets for osteosarcoma identified by proteomic approaches.

## 2. Proteomic Approach

Advances in mass spectrometry have allowed the development of gel-based proteomics from the late 1990s to the early 2000s, while mass spectrometric (MS)-based proteomics has become increasingly popular during the last decade [17].

In the gel-based proteomic approach, the proteome profile is evaluated by protein spot numbers and the intensities detected on the two-dimensional gel electrophoresis (2-DE) gels [18,19,20]. The role of mass spectrometry is mainly focused on protein identification from the 2-DE spot of interest. Matrix-assisted laser desorption ionization-time of flight (MALDI-TOF) has been applied to identify proteins based on a specific pattern of tryptic peptide masses known as the peptide mass fingerprint (PMF). Later, liquid chromatography-tandem mass spectrometry (LC-MS/MS) became the method of choice to identify proteins from the 2-DE spots by inducing fragmentation of tryptic peptides and assembling those fragmented ions into amino acid sequences of the identified peptides and inferred proteins. Differential in-gel electrophoresis (DIGE), which directly labels proteins with fluorescent dyes prior to 2-DE, improves the confidence of in-gel protein spot matching, comparison, and quantification [21]. The advantages of gel-based proteomics include: (i) the differential expression analysis is performed at the protein level (i.e., the different intensities of protein spots between two comparing conditions); and (ii) the visually observable post-translational modifications (PTMs) of proteins (e.g., the shifting of isoelectric points and molecular weights due to protein phosphorylation and glycosylation, respectively) [22,23]. However, gel-based proteomics has decreased in popularity due to several disadvantages, including the low-throughput nature of the workflow and equipment for protein identification and quantitation, the tremendous workload of the entire experiment, and a relatively long time spent to complete each proteomic project.

MS-based proteomics addresses the limitations of gel-based proteomics, increases the depth of proteome coverage, and reduces the workload and time spent per proteomic experiment [20]. Typically, a nanoflow high-performance liquid chromatography with a C18 reverse phase column is utilized to separate tryptic digested peptides of the complex sample before injecting them into the electrospray, which converts peptides in the liquid phase into precursor ions in the gas phase [20]. The mass analyzer then separates precursor ions (and their fragmented ions from collision-induced dissociation) based on the mass-to-charge (*m*/*z*) ratios in electromagnetic fields before hitting the ion detector. Different mass analyzers utilize different methods to measure ions. The combinations of various mass analyzers into the hybrid MS/MS, i.e., Quadrupole-Time-of-Flight (Q-ToF), Trapped Ion Mobility Spectrometry (TIMS)-Q-ToF, Linear Ion Trap-Orbitrap, or Quadrupole-Linear Ion Trap-Orbitrap, have improved the overall performance and become indispensable instruments for modern proteomics [20,24].

MS-based proteomics can be classified into two categories: discovery (or shotgun) proteomics and targeted proteomics. Discovery proteomics utilizes data-dependent acquisition (DDA) to globally identify and quantify as many proteins as possible in biological samples unbiasedly. However, DDA has inherent challenges due to its stochastic and competitive features, resulting in substantial numbers of missing values, particularly for low-abundance peptide ions [20,25]. These circumstances can be addressed by the matching-between-runs option and by optimizing the ion times of individual ions, for instance, with BoxCar, which can reduce the missing values [26]. On the other hand, multiplex isobaric labeling improves the intensities of individual precursor ions while giving a precise mass spectra-based relative quantification with a trade-off of expensive reagents [27]. Despite these attempts, it is impossible to eliminate variability between runs in discovery proteomics [20,25,28]. Targeted proteomics can be performed by three methods, including Multiple Reaction Monitoring (MRM), Parallel Reaction Monitoring (PRM), and Sequential Window Acquisition of all THeoretical fragment ions (SWATH). Contrary to discovery proteomics, targeted proteomics requires knowledge a priori to detect and quantify protein targets in complex samples with a higher reproducibility [20]. MRM and PRM usually need stable isotope tagged-peptide standards for highly accurate quantification of 10s–100s targeted proteins, while SWATH is the targeted label-free method with an ability to quantify 100s–1000s targeted proteins in a single run [20,29,30,31].

MS-based proteomics can profile and map the PTM of proteins (i.e., phosphorylation and ubiquitination) to recognize alterations in protein functions and diverse regulatory mechanisms of cells in biology and disease states. Usually, PTM-proteomics require a pre-requisite step of PTM-peptide enrichment before proteome identification and quantification by MS/MS analysis. Immobilized metal affinity chromatography (IMAC) and antibody capture beads are commonly applied to enrich phosphopeptides [32,33], while the Lys-ϵ-Gly-Gly (diGly) motif antibody-based enrichment has been successfully used for ubiquitinated proteomics [20,34]. Since these reagents and devices are commercially available, PTM-proteomic data acquisition is not a big concern. Nonetheless, data analysis is still a rate-limiting step of the PTM-proteomic project. It should be noted that no single proteomic approach is perfect but rather complementary to each other. Choosing the right proteomic approach that fits the purpose is critical in the initial phase of each proteomic project.

## 3. Proteomics in Osteosarcoma

In recent decades, proteomic studies in osteosarcoma began with a gel-based approach, including 2-DE and two-dimensional difference gel electrophoresis (2D-DIGE), followed by LC-MS/MS. The 2D-DIGE enables us to compare two or three protein samples simultaneously on the same gel, overcoming the irreproducibility limitation of the 2-DE experiment [35]. Later studies of the osteosarcoma proteome were performed through MS-based technology. The advancement of unbiased MS-based proteomics enables quantitative profiling of proteins, PTM mapping, and protein interactions. Figure 1 illustrates the workflow of the proteomic approach in osteosarcoma.

Several osteosarcoma proteomic studies have been conducted from different input materials, including osteosarcoma tissues, cell lines, patient-derived cells, blood, and formalin-fixed paraffin-embedded (FFPE) tissues. This review provides an overview of osteosarcoma proteomic studies that identify valuable clinical biomarkers and targets for osteosarcoma treatment (Table 1).

### 3.1. Osteosarcoma Cell Lines

With a limitation in osteosarcoma tissues, cell lines are one of the crucial models representing osteosarcoma-related protein profiles. Furthermore, their rapid expansion, homogeneity, and reproducibility make cell lines an attractive source for proteomics studies [36,37].

Plasma membrane and global proteome profiles have been conducted using osteosarcoma cell lines as a model. Two studies used 2-DE and LC-MS/MS [38] or isobaric tags for relative and absolute quantification (iTRAQ) technology with LC-MS/MS [39] to compare the plasma membrane proteome of an osteosarcoma cell line (MG-63) vs. an osteoblastic cell line (hFOB1.19). Zhang et al. discovered 68 differentially expressed proteins in osteosarcoma cells [39]. Among them, 69.8% of the identified proteins, including CD151, are located on the plasma membranes. The overexpression of CD151 is further confirmed by immunohistochemistry in osteosarcoma tissues. The other study successfully identified seven differentially expressed proteins in osteosarcoma cells [38]. Among these, the immunohistochemistry staining experiment supports the result from MS data that the N-Myc downstream regulated 1 (NDRG1) is overexpressed in osteosarcoma tissue compared to adjacent non-tumor tissues.

Posthumadeboer et al. conducted a surface proteomic study of several osteosarcoma cell lines vs. human primary osteoblastic cells using 1D and an LC-MS/MS approach [40]. They identified a total of 2841 proteins, 156 of which were surface proteins significantly overexpressed on osteosarcoma cells. The localization of ephrin type-A receptor 2 (EPHA2) was further validated using FACS analysis, which confirms the MS data of EPHA2 overexpression on the surface of osteosarcoma cells. They also demonstrated a significant correlation between a high level of EPHA2 and a poor outcome in osteosarcoma patients.

Global proteome profiles of osteosarcoma cell lines were investigated using 2-DE with MALDI-TOF/TOF [41]. Gemoll et al. identified differentially expressed proteins in osteosarcoma cell lines or pulmonary metastatic cell lines derived from osteosarcoma vs. fetal osteoblastic cell lines. For 17 identified proteins, they found 13 upregulated and 4 downregulated proteins in osteosarcoma and metastatic cells compared to osteoblastic cells. Among all the differentially expressed proteins, cathepsin D is the most promising candidate biomarker. The overexpression of cathepsin D was further confirmed by subsequent experiments in osteosarcoma cell culture and tissue samples.

### 3.2. Patient-Derived Cells

Patient-derived cells have been employed for drug screening and functional testing, bypassing the barrier of drug response prediction based solely on the genetic background of the tumor [42]. Our previous study demonstrates that patient-derived osteosarcoma cells retain their chemo-responsive phenotype, reflecting tumor tissue responses [43]. However, only a few publications investigate the proteome of patient-derived osteosarcoma cells.

The study of Folio et al. compares the proteome profiles of five matched patient-derived normal and osteosarcoma cells using 2D-DIGE and LC-MS/MS [44]. Among 16 identified proteins, alpha-crystallin B chain (CRYAB) and ezrin (EZR1) were further validated using immunohistochemistry of tissue microarrays of paired osteosarcoma and normal tissues. The results confirmed the higher expression of CRYAB and EZR1 in osteosarcoma.

Pruksakorn et al. performed proteomics in osteosarcoma patient-derived cells and osteoblasts derived from bone grafts of non-cancer donors using 2-DE with LC-MS/MS [45]. A total of seven protein spots that are significantly upregulated or downregulated in osteosarcoma were identified by LC-MS/MS. Among these proteins, immunohistochemistry confirmed the overexpression of KH-type splicing regulatory protein (KSRP) in osteosarcoma biopsy samples.

### 3.3. Blood Samples

A sampling of body fluids, mainly plasma or serum, is a non-invasive approach to measure biomarkers for diagnosis, monitoring disease progression along the treatment, as well as stratifying patients for the targeted treatment [46]. Several serum biomarkers identified from MS-based proteomics are potentially applied as prognostic predictions and monitoring biomarkers due to their altered levels during different stages of osteosarcoma treatments.

The serum proteome study of osteosarcoma patients compared to healthy volunteers using 2D-DIGE and MALDI-TOF MS shows a high level of serum amyloid protein A (SAA) in osteosarcoma patients [47]. Immunoblotting and ELISA analysis confirmed increased SAA levels in the serum of osteosarcoma patients. The levels of SAA were decreased after the treatment of a combination of cisplatin and methotrexate and further reduced after surgical removal of the tumor. Of note, SAA levels were raised significantly in relapsed patients.

The proteome profile of plasma from osteosarcoma patients who are sensitive to chemotherapy (good responders) and those who are resistant (poor responders) was examined using surface-enhanced laser desorption/ionization-time of flight mass spectrometry (SELDI-TOF MS) [48]. SAA is also a biomarker for predicting chemo-responses, in which higher expression of SAA was detected in good responders in both pre- and post-chemotherapy plasma of osteosarcoma patients.

More serum proteomics studies investigating the serum proteome of osteosarcoma and healthy donors reported consistent candidate biomarkers. A total of 58 differentially expressed proteins were detected using 2D-DIGE combined with MALDI-TOF MS [49]. The expression of gelsolin was downregulated in osteosarcoma serum compared to healthy controls, in which ELISA and immunoblotting analysis confirmed the results.

Ab-Rahim et al. compare serum proteomes of healthy controls to pre- or post-chemotherapy serum of metastatic osteosarcoma patients and pre-chemotherapy vs. post-chemotherapy using iTRAQ analysis [50]. Consistent with the results from Jin et al., gelsolin was downregulated in pre-chemotherapy serum compared to controls. Furthermore, gelsolin and vascular adhesion molecule-1 (VCAM1) were increased in post-chemotherapy compared to pre-chemotherapy samples.

Osteosarcoma serum biomarkers have been discovered through integrating proteome profiles and gene microarray analysis [51]. The study used a SELDI-TOF-MS protein chip to profile the proteome of serum samples of osteosarcoma patients and healthy controls. In addition, a gene microarray analysis was conducted to identify differentially expressed genes between osteosarcoma cell lines and an osteoblastic cell line. Overexpression of cytochrome C1 (CYC-1) was validated as a candidate biomarker for early osteosarcoma diagnosis.

### 3.4. Tissue Samples

Cancer tissue is an important biological specimen for researching disease etiology at the genomic and proteomic levels. Several proteome profiling of osteosarcoma tissues has been performed.

Li et al. used 2-DE in combination with MALDI-TOF MS to compare benign bone tumors to osteosarcoma tissues [52]. The researchers identified 18 proteins from 30 differentially expressed protein spots. Among all identified proteins, tubulin-a1c (TUBA1C) and zinc finger protein 133 (ZNF 133) were further confirmed for their overexpression in osteosarcoma tissues using immunohistochemistry.

To find biomarkers for predictive chemotherapy response, two studies are used 2D-DIGE and LC-MS/MS [53,54]. Although the number of differentially identified proteins is different, they found that peroxiredoxin 2 (PRDX2) is increased in osteosarcoma patients who poorly respond to chemotherapy compared to good responders.

Our previous proteomic study used 2-DE and LC-MS/MS to perform proteome profiling of soft callus tissues and osteosarcoma tissues [55]. Among 33 differentially protein spots, key mediators in the unfolded protein response (UPR) pathway, including 78 kDa glucose-related protein (GRP78), endoplasmin (GRP94), calreticulin (ERp60), and prelamin-A/C, are upregulated in osteosarcoma tissues.

### 3.5. Formalin-Fixed Paraffin-Embedded (FFPE) Tissues

In recent years, proteomics technology has advanced dramatically. It is now capable of highly comprehensive and quantitative peptide profiling of archived biological materials, including Formalin-Fixed Paraffin-Embedded (FFPE) cancer tissues [56,57].

The proteomics study from the Rao group identified differentially expressed proteins in adult osteosarcoma [58]. The protein expressions of desmoid tumor and adult osteosarcoma FFPE tissues were examined using label-free protein quantification and LC-MS/MS, in which up to 680 unique proteins can be successfully identified. Clusterin and heat shock protein 90 (HSP90) levels were further examined in osteosarcoma tissues with a differential intensity of immunohistochemical staining.

### 3.6. Candidate Biomarkers

From proteomic studies of osteosarcoma, we emphasize the identified proteins with validation experiments or functional studies as potential biomarkers and therapeutic targets for osteosarcoma treatment and management.

#### 3.6.1. Ezrin

Ezrin, a member of the ezrin/radixin/moesin (ERM) protein family, acts mainly as a cross-linker between the actin cytoskeleton and membrane proteins or phospholipids in the cell membranes [59]. It also mediates G-protein-related proteins and signal transduction in the plasma membrane. Ezrin plays a crucial role in cancer metastasis [60]. The overexpression of ezrin is significantly associated with poor clinical outcomes in many cancers.

In osteosarcoma, ezrin expression has been associated with lung metastasis in vivo [61,62], and inhibition of ezrin reduces the metastasis [63,64]. High ezrin expression has been linked to a poor prognosis in osteosarcoma [65,66,67]. A meta-analysis reveals that patients with positive ezrin have shorter overall survival than negative ezrin patients, suggesting that ezrin may be a potential prognostic marker in osteosarcoma patients [68,69]. Furthermore, the elevated ezrin expression in circulating tumor cells (CTCs) correlates with distant metastases [70]. Ezrin can be a valuable biomarker for prognosis and therapeutic target in osteosarcoma.

The mechanism of ezrin-mediated metastasis in osteosarcoma involves kinase pathways. A high level of ezrin induces the phosphorylation of both Akt and p44/42 MAPK [65]. Ezrin-induced metastasis is mediated by the MAPK pathway [65] and the mTOR/S6K1/4E-BP1 pathway [71].

MiR-96, miR144, miR-150, miR-183, and miR-211 inhibit cell proliferation, invasion, migration, and tumor formation by targeting ezrin [72,73,74,75,76]. Furthermore, the natural compound, Baicalein, effectively suppresses the invasion and migration of osteosarcoma cell lines through upregulation of miR-183 and downregulates the mRNA and protein level of ezrin [77].

#### 3.6.2. CRYAB

αB-crystallin (CRYAB) is a human small heat-shock protein (sHsp) that is involved in tumorigenesis in several types of cancer, including osteosarcoma [78]. The elevated expression of CRYAB was detected in osteosarcoma tissues and cell lines. High levels of CRYAB are associated with poor prognosis in osteosarcoma [78]. CRYAB induced lung metastasis in vivo, in which inhibition of CRYAB reduced tumor size through the MEK/ERK signaling pathway.

Overexpression of CRYAB expression in osteosarcoma is regulated through krüppel-like factor 4 (KLF4), a zinc-finger transcription factor [79]. A binding of KLF4 with a promoter of CRYAB transactivated CRYAB expression in osteosarcoma cells. Expression of both KLF4 and CRYAB was higher in osteosarcoma tissues compared to normal bone tissues. CRYAB increased the migration and tumor formation of osteosarcoma cells.

The role of miR-491 has been investigated in the metastasis and chemoresistance of osteosarcoma [80]. Levels of serum miR-491 are reduced in osteosarcoma patients compared to healthy donors. The decrease in serum miR-491 levels is linked to increased metastasis, poor chemo-responses, and a shorter survival rate. In both in vitro and in vivo studies, miR-491 overexpression potentially inhibited lung metastasis and increased the sensitivity of osteosarcoma to cisplatin treatment by directly targeting CRYAB, which was identified as a direct target of miR-491. The restoration of miR-491 expression results in an inhibition of CRYAB expression.

#### 3.6.3. CD151

CD151, a tetraspanin family member, mediates cell–cell and cell–extracellular matrix interactions [81]. It is involved in tumor development and metastasis [82]. CD151 was shown to be highly expressed in osteosarcoma cell metastasis. A knockdown of CD151 inhibits osteosarcoma lung metastasis in vivo via the GSK-3/-catenin/MMP9 pathway [72].

The silencing of CD151 also reduces Akt and P38 phosphorylation, which are critical regulators in cancer cell motility and migration [81,83,84]. Furthermore, osteosarcoma patients with elevated CD151 expression have poor overall survival [70].

#### 3.6.4. EPHA2

Ephrin type-A receptor 2 (EPHA2), a tyrosine kinase receptor, is a significant oncoprotein involved in self-renewal, angiogenesis, and metastasis in various solid malignancies [85]. Osteosarcoma patients with EPHA2-positive tumors have a worse survival rate [40].

Targeting EPHA2 is promising for the treatment of osteosarcoma. Chimeric Antigen Receptor-modified T (CAR T) cells that target EPHA2 have significant anti-tumor efficacy in vitro and can eradicate osteosarcoma tumors in vivo [86]. Systemically injected CAR T cells targeting EPHA2 can traffic to and eliminate tumor deposits in murine livers and lungs in an aggressive metastatic osteosarcoma [86]. Furthermore, the combination of pazopanib and trametinib targets PI3K/Akt and MEK/ERK pathways, inhibiting tumor growth through a decreased expression of EPHA2 [87].

#### 3.6.5. Cathepsin D

Cathepsin D (CTSD) is an aspartic protease enzyme in the pepsin family [88]. It plays a vital role in various steps of cancer progression and development, such as metastasis, invasion, migration, angiogenesis, and cell proliferation [89,90].

In osteosarcoma, cathepsin D might play an essential role in metastasis and chemoresistance. The results from immunohistochemistry of tissue microarrays demonstrated the overexpression of cathepsin D in lung metastases and osteosarcoma tissues compared to normal bone tissues [41]. Expression of cathepsin D at both mRNA and protein levels was increased in highly metastatic osteosarcoma cells compared to non-metastatic cells [91]. Overexpression of cathepsin D was also observed in osteosarcoma cells cultured in a spheroid form, where the osteosarcoma cells developed doxorubicin resistance compared to osteosarcoma cells cultured in the monolayer [92].

Even though no functional study of cathepsin D has been performed in osteosarcoma, our earlier work in proteome data mining suggests that cathepsin D is a potential biomarker and target for osteosarcoma treatment [93]. We identified candidate therapeutic targets by cross-referencing proteins, which are upregulated in osteosarcoma compared to osteoblastoma, in chemoresistance compared to chemo-sensitive, and in metastatic compared to non-metastatic patients, with identifiers of targets of FDA-approved agents and chemical inhibitors. Cathepsin D is the only target of a small molecule inhibitor that is highly expressed in all investigated conditions.

#### 3.6.6. GRP78

GRP78 (BiP or HSP70 family protein 5) is a chaperone residing in the endoplasmic reticulum and plays a crucial role in the regulation of the unfolded protein response (UPR) pathway [94]. The proteomic study shows an activation of the UPR in osteosarcoma tissues compared to soft callus tissues, resulting in the overexpression of GRP78 and other chaperone genes in the ATF6 arm of the UPR [55].

GRP78 is upregulated in the osteosarcoma tissues derived from patients who respond poorly to chemotherapy compared to good responders [55]. In addition, the overexpression of GRP78 was found at the mRNA and protein levels in metastatic osteosarcoma tissues compared to non-metastases in immunofluorescence staining, RT-PCR, and western blot analysis [95].

Both doxorubicin and cisplatin treatments can trigger the UPR and GRP78 overexpression in osteosarcoma cells. The expression level of GPR78 is increased with the activation of AKT activity after the treatment of doxorubicin (DOX) [96]. This activation causes a high level of P-glycoprotein (P-gp), a key player in acquired multidrug resistance (MDR) in osteosarcoma, particularly in the DOX-resistant sublines. The study further demonstrates an association of GRP78, Akt, and P-gp under DOX stimulation, in which GRP78 can promote Akt phosphorylation and enhance Akt-mediated P-gp expression. In concordance, GRP78 silencing suppresses the expression of P-gp in both parental osteosarcoma cells and the DOX-resistant sublines.

In response to cisplatin treatment of osteosarcoma cells, the UPR is activated via an enhancement of the NF-κB signaling pathway, in which the levels of GRP78 and CHOP are significantly increased [97].

The significance of GRP78 in the resistance of osteosarcomas to bortezomib (BTZ), the proteasome inhibitor, is also investigated. The study shows a loss of activating transcription factor 4 (ATF4), which is a downstream effector of pancreatic EIF2-α kinase (PERK) and a key regulator of the cellular stress response in BTZ resistance of osteosarcoma [98]. The underlining mechanism of ATF4 downregulation involves a positive feedback loop of GRP78 and RET, a receptor tyrosine kinase (RTK) member. In this way, ATF4 interacts with RET to recruit it to proteasomal degradation. In turn, a loss of ATF4 maintains the RET level and causes BTZ resistance. The binding of GRP78 and RET, on the other hand, interferes with the ATF4/RET interaction and promotes RET stabilization.

Interestingly, a recent study investigates the effects of a natural compound that might have an anti-GRP78 function in osteosarcoma cells. The treatment of kuanoniamine C, isolated from *Oceanappia* sp., causes GRP78 mRNA degradation that induces osteosarcoma cell death [99]. The p53 signaling pathway regulates this GRP78 inhibition-induced cell death. The combination of kuanoniamine C and BTZ effectively reduces the expression of GRP78, which is important for stimulating BTZ-induced cell death.

#### 3.6.7. HSP90

HSP90 is a molecular chaperone highly conserved from bacteria to humans [100]. It controls the maturation and folding of many oncogenic proteins, making it an essential mediator for the survival of cancer cells. HSP90 is involved in a wide range of tumorigenic processes of osteosarcoma, including cell proliferation, autophagy, apoptosis, and metastasis [101,102,103,104]. Compared to a desmoid tumor, the proteome profile of osteosarcoma tissue from the FFPE archive demonstrated an overexpression of HSP90 in osteosarcoma [58]. The expression of HSP90 was further validated in 16 post-chemotherapy tissue microarrays using immunohistochemistry. The results showed an upregulation of HSP90 in osteosarcoma tissues compared to benign tumors. The functions of HSP90 in osteosarcoma tumorigenesis have been investigated through the treatment of various HSP90 inhibitors. Most studies demonstrated suppression of AKT signaling and related crosstalk pathways following HSP90 inhibition [101,102,103,105].

Geldanamycin (GA), a benzoquinone ansamycin antibiotic, is the first to be established as an HSP90 inhibitor [106]. GA induces autophagy and apoptosis in osteosarcoma cells by regulating the Akt/mTOR signaling pathway [102]. The treatment of GA decreases the phosphorylation of downstream effectors of the Akt/mTOR pathways, including phosphor-mTOR, phosphor-p70S6K, and phosphor-4E-BP1. The clinical use of GA is limited due to its high hepatotoxicity, poor solubility, and metabolic instability.

The geldanamycin derivative, 17-AAG, is the first HSP90 inhibitor that has entered clinical trials for cancer treatment [107]. Inhibition of HSP90 using 17-AAG significantly reduces osteosarcoma cell growth and induces apoptosis by decreasing the expression of Runx2, a regulator of the differentiation of osteoblasts [103]. HSP90 transcriptionally regulates this Runx2 downregulation through the Akt/GSK-3β/β-catenin axis.

PF4942847 is a novel synthetic HSP90 inhibitor. It is a nonquinone-based compound designed to improve pharmacologic properties and toxicity to overcome a limitation of the clinical use of the first generation of HSP90 inhibitors [108]. PF4942847 inhibits osteosarcoma cell growth and increases apoptosis by suppressing Akt, p-ERK, c-Met, and c-RAF1 [101]. In an in vivo study, PF4942847 inhibits osteosarcoma tumor growth by reducing cell proliferation and an induction of apoptosis. HSP90 inhibition also decreases lung metastasis by reducing the cascade of c-Met, MMP9, and focal adhesion kinases (FAK).

A recent study has shown that HSP90 regulates threonine and tyrosine protein kinase (TTK) by directly interacting with TTK and preventing the proteasome degradation of TKK, resulting in the accumulation of TTK in osteosarcoma cells [104]. The overexpression of TTK induces a rapid process of the cell cycle and significantly enhances osteosarcoma cell proliferation. Inhibition of HSP90 using GA and 17-AAG can restore G2/M cell cycle arrest and an accumulation of aneuploid cells, suppressing osteosarcoma cell proliferation. The treatment of HSP90 inhibitors also causes a reduction in tumor growth in mice carrying TTK-overexpressed osteosarcoma.

This evidence all suggests the potential use of HSP90 inhibitors for the treatment of osteosarcoma. A Phase I clinical trial of 17-AAG was conducted in pediatric cancer patients, including osteosarcoma, neuroblastoma, Ewing’s family tumors, and desmoplastic small round cell tumors [109]. The study reported a safe dose of 17-AAG in pediatric patients with refractory solid tumors, with a precaution in patients with advanced pulmonary diseases.

**Table 1 ijms-23-09741-t001:** Overview of proteomic studies in osteosarcoma (OS).

Sample Type	Techniques	Sample Information		Number of Proteins	Candidate Biomarker	References
		Control Group	Disease Group			
**Patient-derived cell and cell line**	2D-DIGE and LC-ESI-MS/MS (Q-TOF)	Osteoblastic cell	Primary OS tumor cell	56 differential protein spots 16 proteins identified	Ezrin (EZR) ↑ and alpha Crystallin B chain (CRYAB) ↑	[44]
	iTRAQ, LC-MS/MS (Q-TOF)	hFOB	MG-63	342 proteins identified 68 differentially expressed proteins	CD151 ↑	[39]
	2-DE and LC-ESI-MS/MS (HCT ion trap)	hFOB	MG-63	259 protein spots (hFOB) 222 protein spots (MG-63) 13 differentially expressed protein spots 7 protein spots identified	N-Myc Downstream Regulated 1 (NDRG1) ↑	[38]
	1D and LC-MS/MS (LTQ-FT)	Human primary osteoblast	OS cell lines	2841 proteins identified 684 significant protein 151 surface proteins 43 selected proteins	Ephrin type-A receptor 2 (EPHA2) ↑	[40]
	2-DE and MALDI-TOF MS	Fetal osteoblastic cell	OS cell line and pulmonary metastases derived from OS	~1114–1791 protein spots 34 differentially expressed protein spots 17 protein spots identified	Cathepsin D (CTSD) ↑	[41]
	2-DE and LC-MS/MS (ion trap)	Osteoblasts of cancellous bone	OS primary cell	~415 protein spots (Osteoblast) ~348 protein spots (OS cells) 257 protein spots matched 33 differentially expressed protein spots 7 proteins identified	KH-type splicing regulatory protein (KSRP) ↑	[45]
**Plasma or serum**	2D-DIGE and MALDI-TOF MS	Healthy volunteer	Osteosarcoma patient	1050–1100 protein spots 58 differentially expressed protein spots 43 protein spots identified	Serum amyloid protein A (SAA) ↑	[47]
	SELDI-TOF MS	Healthy volunteer	Osteosarcoma patient	96 differentially expressed protein peaks 6 significantly expressed protein peaks	Cytochrome C1 (CYC-1) ↑	[51]
	SELDI-TOF MS	Pre-chemotherapy (Good responders) Post-chemotherapy (Good responders)	Pre-chemotherapy (Poor responders) Post-chemotherapy (Poor responders)	783 protein peaks identified 56 protein peaks identified in pre-treatment group 65 protein peaks identified in post-treatment group	Serum amyloid protein A (SAA) ↓	[48]
	2D-DIGE and MALDI-TOF MS	Healthy volunteer	Osteosarcoma patient	1050–1100 protein spots 58 differentially expressed protein spots 43 protein spots identified	Gelsolin ↓	[49]
	iTRAQ, LC-MS/MS (Triple TOF 5600)	Pre-chemotherapy with metastatic OS patient	Post-chemotherapy with metastatic OS patient	217 proteins identified and quantified 57 differentially expressed proteins	Gelsolin ↑ and vascular adhesion molecule-1 (VCAM-1) ↑	[50]
**Tissue**	2-DE and MALDI-TOF MS	Benign tumor of bone (osteoblastoma)	Osteosarcoma	~1270 protein spots detected (Osteoblastoma) ~1386 protein spots detected (OS) 30 differential protein spots 18 proteins identified	Zinc finger protein 133 (ZNF 133) ↑ and tubulin-a1c (TUBA1C) ↑	[52]
	2D-DIGE and LC-nanoES-MS/MS (LTQ linear ion trap)	Chemonaive biopsy; Good responder	Chemonaive biopsy; Poor responder	2250 protein spots detected 55 differential protein spots identified	Peroxiredoxin 2 (PRDX2) ↑	[53]
	2D-DIGE and LC-nanoES-MS/MS (LTQ Oribitrap)	Chemonaive biopsy; Good responder	Chemonaive biopsy; Poor responder	3494 protein spots detected 33 differential protein spots identified	Peroxiredoxin 2 (PRDX2) ↑	[54]
	2-DE and LC-ESI-MS/MS (Q-TOF)	Normal soft tissue callus	Osteosarcoma	329 protein spots matched 32 differential protein spots identified	78 kDa glucose-related protein (GRP78), endoplasmin (GRP94) ↑, calreticulin (ERp60) ↑ and prelamin-A/C ↑	[55]
**FFPE**	LC-MS/MS (LTQ ion trap)	Desmoid tumor	Osteosarcoma	~680 unique protein identified	Clusterin ↑ and heat shock protein 90 (HSP90) ↑	[58]

↑ and ↓ indicate increased and decreased expression of the identified proteins in the disease group compared to the control group, respectively.

## 4. Challenges and Future Perspectives

Osteosarcoma is a deadly bone cancer that has shown no improvement in patient outcomes for decades. The lack of potential biomarkers to stratify individuals who do not respond to chemotherapy and inefficient therapy for this group of patients are major challenges in osteosarcoma treatment. The identification of chemoresistance-related characteristics can be used not only as a novel biomarker but also as a therapeutic target for osteosarcoma patients. With the advancement of current OMICs technology and the integration of multiple OMICs analyses, we now have a better knowledge of the genetic, transcriptomic, and epigenetic modifications that lead to individual drug metabolism and response. Proteomics is now being used to validate the expression of specific genes involved in drug resistance to uncover novel biomarkers that might be used to identify potential therapeutic targets. Figure 2 illustrates an example of the use of novel MS-based technology to accelerate biomarker discovery and therapeutic target identification in chemoresistant osteosarcoma.

Through tissue biopsy, a proteomics study of osteosarcoma identifies candidate biomarkers for drug sensitivity that may indicate a pre-existing resistant mechanism. However, accumulating evidence now confirms the existence of an acquired resistance mechanism after chemotherapy treatment [110]. To understand this drug-induced resistance, it is essential to investigate the molecular profiling of the residual cells, the most advanced cancer cells, before overt metastasis. The residual tumor contains necrotic tissues that are a histological response to chemotherapy treatment. Therefore, for proteomics and most of the OMICs analysis, it is necessary to discard these necrotic tissues to investigate viable cells specifically. Microdissection is a technique we can apply for removing necrotic tissues and for spatial proteomic analysis [111]. The most challenging aspect of implementing microdissection is the sensitivity of downstream proteomics analysis, so nano- or micro-proteomics were introduced. Most of the proteomics studies of micro-dissected tissues are performed on quadrupole orbitrap mass spectrometers and the trapped ion mobility spectrometry (TIMS) technology due to their high sensitivity [112].

Osteosarcoma has mostly been diagnosed and monitored via tissue biopsy and imaging. Tissue biopsy, by its procedure, is an invasive process that is not always accessible and limits repeated samplings not possible for long-term monitoring of disease progression. Liquid biopsy is an alternative, effective, non-invasive approach to overcome these limitations.

Even though the primary tumor is completely eradicated after resection, most osteosarcoma patients who do not respond to treatment develop distant metastases. This scenario indicates an ineffective systemic control of the disease. Circulating tumor cells (CTCs) are disseminated from the primary tumor, enter the bloodstream, and grow at the distant organ. CTCs are a critical intermediate between the original tumor and the metastatic site, representing the disease’s aggressiveness [113]. Detection and characterization of CTCs are highly promising approaches for indicating drug responses, long-term surveillance, and monitoring patients’ drug responsiveness in clinical trials.

CTCs are rare cells, with contamination of white blood cells (WBCs) of approximately 1 CTC in 1 million WBCs per milliliter of peripheral blood [114]. To minimize WBC contamination, various enrichment techniques, mainly positive selection of CTC markers or adverse selection of WBC depletion, have been used [115]. However, most CTC detection methods are still hampered by significant contaminations or, on the other hand, with a higher purity of CTCs but the loss of a substantial fraction of CTCs during enrichment processes.

MALDI-TOF mass spectrometry has been used for bacterial chemotaxonomy analysis in which peptide profiles generated from the MS effectively distinguish different species of microorganisms [116,117]. The MALDI-TOF MS is also used to stratify different phenotypes of mammalian cells, for instance, the identification of “intact cells” or “whole cells” in the differentiation of glial cells and toxic effects in established cell lines [118,119]. Recently, the use of main spectra profiles (MSP) generated from MALDI-TOF MS has been performed in cancer cell line spiking experiments; the results showed that MALDI-TOF MS was optimized to detect CTCs as small as six cells in 5000 WBCs [120]. Due to the high turnaround time, cost-effective, and clinical-friendly workflow, the MALDI-TOF MS is a promising tool for diagnosis, prognosis prediction, and long-term surveillance of patients.

Protein phosphorylation is a significant PTM that modulates several cellular processes both directly and indirectly [121]. Through the action of protein kinases, phosphate groups are added to serine, threonine, or tyrosine residues to transiently alter protein properties, including their enzymatic activities, the interaction with binding partners, their localization and conformations, or to target them to degradation [121,122]. A large number of kinases, particularly tyrosine kinases, have been extensively studied in cancer biology [123]. Kinase inhibitors are now one of the most used for treating various cancers. Therefore, to identify signaling cascades that are significant consequences of aberrant phosphorylation modifications of proteins, phosphoproteomics is an attractive tool for novel drug discovery.

However, there are several challenges in the phosphoproteomics analysis of cancer tissues. First, the levels of phosphorylated proteins in cells are relatively low in both physiological and pathological conditions, in which phosphotyrosines represent only 1–2% of the global phosphorylations of proteins [124]. Second, the quantity of clinical tissues, particularly biopsy samples, is limited for downstream phosphoproteomics. To overcome these limitations, many efforts have been undertaken to improve both mass spectrometry and phospho-peptide enrichment techniques.

An advancement in the phosphoproteomic pipeline now allows the identification of nearly complete proteomes with more than 12,000 proteins and 10,000 PTM sites in clinical samples [125]. Enrichment methods have been developed for global and specific phosphopeptide enrichments, including the use of metal-based affinity chromatography (IMAC and metal oxide affinity chromatography; MOAC) [126,127] and antibody-based immunoprecipitation methods [128].

For phosphopeptide quantification, the label-free approach can identify the phosphotyrosine sites from only 1 mg of total protein input. In contrast, the profiling of serine- and threonine-phosphorylated proteins can be performed from 250 to 500 μg of proteins [124]. Together with software development, for example, Skyline, MaxLFQ, and MaxQuant, label-free phosphoproteomics is now more robust and applicable [129]. Furthermore, metabolic labeling, chemical labeling, and isobaric labeling have been developed for phosphoproteomics quantification. iTRAQ technology has been introduced for multiplexing phosphopeptide analysis of up to ten samples in a single LC-MS/MS run [130].

Integrating the analysis of proteomics and phosphoproteomics data is of the utmost importance for biological interpretation. Recently, a web-based tool named piNET has been introduced to the study, interpretation, and visualization of MS-based proteomics data [131]. PiNET is useful for efficiently mapping peptides/PTMs to proteins, integrating meta-data for PTMs, mapping PTMs to modifying enzymes, and providing additional functional annotations. It can also generate high-quality visualizations of PTM networks and protein pathways.

Osteosarcoma is surrounded by a dynamic bone microenvironment composed of a varied spectrum of cell types, including bone cells, stromal cells, vascular cells, immune cells, and mineralized extracellular matrix (ECM) [132]. The tumor microenvironment (TME) has been reported for its crucial role in osteosarcoma growth and metastasis. Much effort has been focused on the discovery of new therapeutic targets that target TME components. Cancer-associated fibroblasts and tumor-associated macrophages, for example, are hot topics in cancer biology and immunotherapy, including osteosarcoma [133]. In this direction, mass cytometry by time-of-flight (CyTOF) has emerged as a powerful tool for profiling tumor heterogeneity and the TME at a single-cell resolution [134]. The use of CyTOF for decoding the osteosarcoma TME should be pursued in the future.

## 5. Conclusions

The lack of novel biomarkers or effective therapy restricts the effective management and treatment of osteosarcoma. Most patients with poor treatment responses develop lung metastases within a few years of their initial diagnosis. Effective therapies, as well as surveillance monitoring of disease progression, are crucial to fighting osteosarcoma. Unlike adult cancer, osteosarcoma has a lower mutation rate. Multiple regulatory levels are implicated in osteosarcoma carcinogenesis, including transcriptional control, protein expression, and PTM. Therefore, an insight into the understanding of osteosarcoma biology, heterogeneity, and tumor microenvironments through integrative multi-omics might lead to a way to completely eradicate osteosarcoma. In this direction, future studies should be focused on the proteome dynamics of osteosarcoma tissues derived from various phases of therapy to improve our understanding of chemoresistant mechanisms, which ultimately lead to new therapeutic targets. Mass spectrometric-based proteomics is an indispensable approach to discovering candidate biomarkers for stratifying high-risk patients and monitoring disease progression in the era of cancer precision medicine.

## Figures and Tables

**Figure 1 ijms-23-09741-f001:**
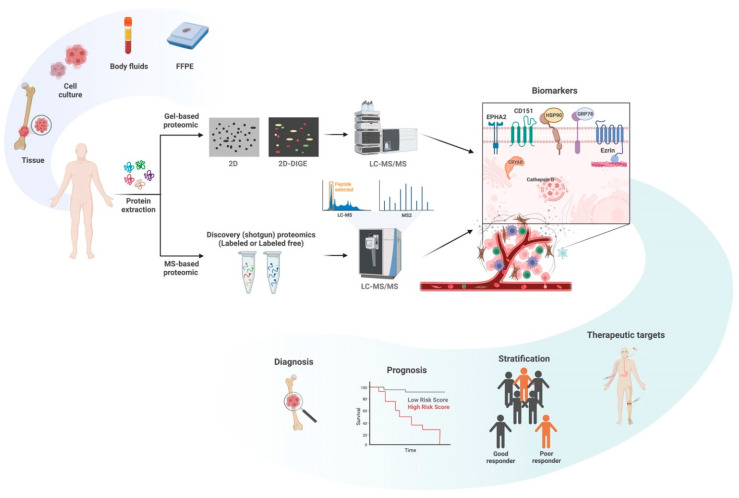
Workflow of the proteomic approach in osteosarcoma.

**Figure 2 ijms-23-09741-f002:**
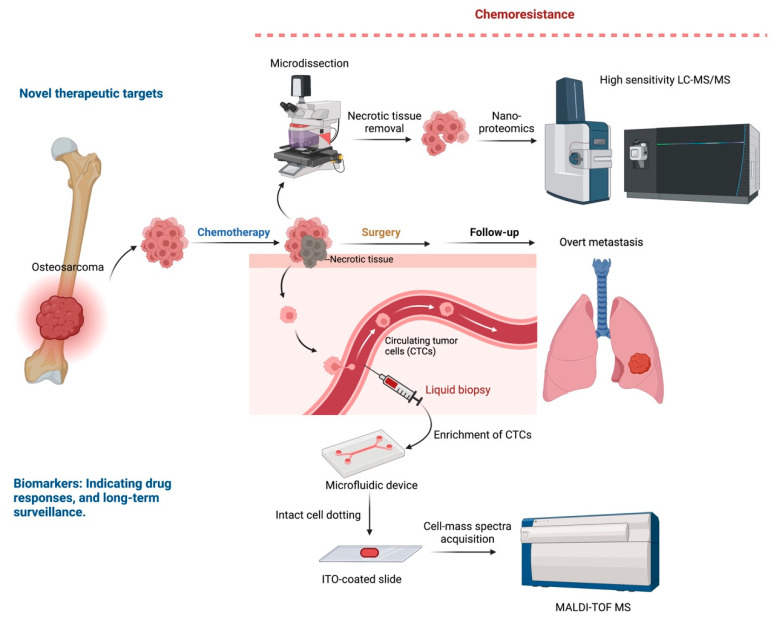
Future perspectives for the use of MS-based technology in novel target and biomarker discovery in osteosarcoma.

## Data Availability

Not applicable.

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
