# Peer review of "Mass Spectrometric-Based Proteomics for Biomarker Discovery in Osteosarcoma: Current Status and Future Direction"

_ijms, 2022, doi:10.3390/ijms23179741_

Round 1
Reviewer 1 Report
This review examines the current state of proteomics and biomarkers in osteosarcoma. Outcomes for patients with osteosarcoma have been stagnant for decades, and it is known that genomic analysis is insufficient for substantial future advancement in this regard. Recent advances in proteomics is expected to help solve this issue. By examining the protein profiles of the current tools used for studying osteosarcoma (cell lines, patient samples, etc.), this review also notes potential biomarkers for outcomes in osteosarcoma. These biomarkers are expected to help stratify patients according to their predicted response to therapy.
Overall, this is a well-written review on the current state of proteomic analysis for osteosarcoma. It is also nicely and appropriately illustrated with informative and attractive figures. Text and references are appropriately extensive but not excessive.
Author Response
Reviewer#1
This review examines the current state of proteomics and biomarkers in osteosarcoma. Outcomes for patients with osteosarcoma have been stagnant for decades, and it is known that genomic analysis is insufficient for substantial future advancement in this regard. Recent advances in proteomics is expected to help solve this issue. By examining the protein profiles of the current tools used for studying osteosarcoma (cell lines, patient samples, etc.), this review also notes potential biomarkers for outcomes in osteosarcoma. These biomarkers are expected to help stratify patients according to their predicted response to therapy.
Overall, this is a well-written review on the current state of proteomic analysis for osteosarcoma. It is also nicely and appropriately illustrated with informative and attractive figures. Text and references are appropriately extensive but not excessive.
Response: We are grateful that the reviewer found this review was well-written and provide useful information to support future research in osteosarcoma proteomics.
Reviewer 2 Report
This paper summarises proteomic analysis devoted to the osteosarcoma and the biomarkers associated with this illness.
Searching and discovery for novel protein biomarker candidates related to severe diseases is currently a challenge and a hot topic for many investigation or clinical laboratories. Study of osteosarcoma based on proteomic analysis also belongs to the actual and important scope of investigation.
In that matter, the review of recent development in this research field does deserve an attention.
I value especially nice figures of a proteomic workflows provided.
The length is not too long, what is advantageous for the pioneer researchers.
The number of references is acceptable so is the English style.
REMARKS
1. In introduction and proteomic approach section, some important references belonged to the current proteomic research is missing. Please, follow the down below upgrade in the 2nd and 3rd paragraph.
“Proteomics has provided complimentary and contrasting data to their genomic counterparts, leading to a comprehensive understanding of the molecular mechanisms underlining the pathology of diseases [https://doi.org/10.1002/pmic.202000318].”
“Discovery proteomics enables large-scale protein identification and the detection of protein dynamics in biological states and pathogenic conditions [https://doi.org/10.1002/pmic.202100198].”
“Advances in mass spectrometry have allowed the development of gel-based prote-65 omics from the late 1990s to the early 2000s, while mass spectrometric (MS)-based prote-66 omics has become increasing popularity during the last decade [https://doi.org/10.1016/j.cca.2020.04.015].“
2. There are many abbreviations used, please summarise them in the abbreviations list.
3. In conclusion, please provide some future aims of the authors in this scope of investigation.
Author Response
Reviewer#2
This paper summarises proteomic analysis devoted to the osteosarcoma and the biomarkers associated with this illness.Searching and discovery for novel protein biomarker candidates related to severe diseases is currently a challenge and a hot topic for many investigation or clinical laboratories. Study of osteosarcoma based on proteomic analysis also belongs to the actual and important scope of investigation. In that matter, the review of recent development in this research field does deserve an attention. I value especially nice figures of a proteomic workflows provided. The length is not too long, what is advantageous for the pioneer researchers. The number of references is acceptable so is the English style.
REMARKS
1. In introduction and proteomic approach section, some important references belonged to the current proteomic research is missing. Please, follow the down below upgrade in the 2ndand 3rd paragraph.
“Proteomics has provided complimentary and contrasting data to their genomic counterparts, leading to a comprehensive understanding of the molecular mechanisms underlining the pathology of diseases [https://doi.org/10.1002/pmic.202000318].”
“Discovery proteomics enables large-scale protein identification and the detection of protein dynamics in biological states and pathogenic conditions [https://doi.org/10.1002/pmic.202100198].”
“Advances in mass spectrometry have allowed the development of gel-based prote-65 omics from the late 1990s to the early 2000s, while mass spectrometric (MS)-based prote-66 omics has become increasing popularity during the last decade [https://doi.org/10.1016/j.cca.2020.04.015].“
Response: We have updated the references, accordingly, as shown in page 2, lines 47, 51, and 68.
2. There are many abbreviations used, please summarise them in the abbreviations list.
Response: We have provided the abbreviation list in page 14, lines 578-649.
3. In conclusion, please provide some future aims of the authors in this scope of investigation.
Response: Thank you very much for your suggestion. We have included our future aims in the conclusion session as follows;
Page 14, lines 572-576:
“… In this direction, future studies should be focused on the proteome dynamics of osteosarcoma tissues derived from various phases of therapy to improve our understanding of chemo-resistant mechanisms, which ultimately lead to new therapeutic targets. Mass spectrometric-based proteomics is an indispensable approach to discover candidate biomarkers for stratifying high-risk patients and monitoring disease progression in the era of cancer precision medicine.”
Reviewer 3 Report
In this review article Sirikaew et al. introduce the application of Mass spectrometric (MS)-based proteomics to osteosarcoma. This review is very unique, innovative, and highly informative, and does not seem to need revision as it stands. This reviewer just recommends one point.
1. Recently, the demand for CyTOF analysis has increased significantly in cancer biology area. It is expected that the application of CyTOF to osteosarcoma will increased in the future. This reviewer recommends the authors describe it shortly in future perspectives.
Author Response
Reviewer#3
In this review article Sirikaew et al. introduce the application of Mass spectrometric (MS)-based proteomics to osteosarcoma. This review is very unique, innovative, and highly informative, and does not seem to need revision as it stands. This reviewer just recommends one point.
1. Recently, the demand for CyTOF analysis has increased significantly in cancer biology area. It is expected that the application of CyTOF to osteosarcoma will increased in the future. This reviewer recommends the authors describe it shortly in future perspectives.
Responses: Thank you very much for your suggestion. We agree that the CyTOF analysis could provide critical information on osteosarcoma biology, particularly the tumor microenvironment. We have added new statements in the future perspective section as follows;
Page 13, lines 545-554:
“Osteosarcoma is surrounded by a dynamic bone microenvironment composed of a varied spectrum of cell types, including bone cells, stromal cells, vascular cells, immune cells, and mineralized extracellular matrix (ECM) [132]. The tumor microenvironment (TME) has been reported for its crucial role in osteosarcoma growth and metastasis. Much effort has been focused on the discovery of new therapeutic targets that target TME components. Cancer-associated fibroblast and tumor-associated macrophages, for example, are hot topics in cancer biology and immunotherapy, including osteosarcoma [133]. In this direction, mass cytometry by time-of-flight (CyTOF) has emerged as a powerful tool for profiling tumor heterogeneity and the TME at a single-cell resolution [134]. The use of CyTOF for decoding the osteosarcoma TME should be pursued in the future.”
Round 2
Reviewer 2 Report
Authors have reacted to the given queries. Thus, I consider publication of the manuscript at current form.